# Concentrations of potentially toxic metals and trace elements in pregnant women and association with birth outcomes: A cross-sectional study in Malawi

**Mphatso Mwapasa**[1]*, **Shanshan Xu**[2], **Bertha Magreta Chakhame**[1,3], **Alfred Maluwa**[4], **Halina Röllin**[5], **Augustine Choko**[6], **Sandra Huber**[7], **Jon Øyvind Odland**[1,4,5]

**1** Department of Public Health and Nursing, Norwegian University of Science and Technology, Trondheim, Norway, **2** Centre for International Health, Department of Global Public Health and Primary Care, University of Bergen, Bergen, Norway, **3** School of Maternal, Neonatal and Reproductive Health, Kamuzu University of Health Sciences, Blantyre, Malawi, **4** Directorate of Research and Outreach, Malawi University of Science and Technology, Thyolo, Malawi, **5** School of Health Systems and Public Health, Faculty of Health Sciences, University of Pretoria, Pretoria, South Africa, **6** Malawi Liverpool Wellcome Trust Clinical Research Programme, Blantyre, Malawi, **7** Department of Laboratory Medicine, University Hospital of North Norway, Tromsø, Norway

* mphomwapasa@gmail.com

**Data Availability Statement:** Dataset for this study is available on the public repository (Github

## Abstract

Potentially toxic metals and trace elements have been used in Malawi for a long time. However, data on exposure to these elements by susceptible groups like pregnant women and its associations with reproductive health outcomes in Malawi and southern hemisphere is limited. We investigated the concentrations of potentially toxic metals as well as trace elements in pregnant women and assessed the relationship between the levels these elements in maternal blood and sociodemographic factors, dietary habits and birth outcomes. Maternal data was collected from 605 pregnant women. Provider administered questionnaire was used to collect data on maternal sociodemographic factors, life style and immediate birth outcomes. Maternal venous blood samples were collected from 506 pregnant women in southern Malawi between August 2020 and July 2021. An inductively coupled plasma mass spectrometry (ICP-MS) technique was used to analyse maternal blood samples for concentrations of arsenic (As), copper (Cu), lead (Pb), mercury (Hg), nickel (Ni), selenium (Se) and zinc (Zn). Maternal age emerged as the primary predictor for Cu ($p = 0.023$), As ($p = 0.034$) and Hg ($p = 0.013$) blood concentrations, followed by area of residence, which had significant impact on Ni ($p = 0.024$) and As ($p < 0.001$) concentrations. High maternal blood concentrations of Ni were associated with increased birth weight ($p = 0.047$), birth length ($p = 0.026$), head circumference ($p = 0.029$) and gestational age ($p = 0.035$). Negative associations were observed between maternal whole blood total arsenic (combining organic and inorganic As) concentration and neonatal birth length ($p = 0.048$) and head circumferences ($p < 0.001$). Similarly, higher maternal blood Pb concentrations were associated with smaller head circumference ($p = 0.002$) and birth weight ($p = 0.016$). This study demonstrates the need to introduce biomonitoring studies in Malawi and countries with similar settings in the global south.

repositoty): https://github.com/MphatsoMwapasa/
Toxic-essential-Metals-Data.git.

**Funding:** This research work is supported by the
Norwegian University of Science and Technology,
(grant number 90380701 to JØO) and the Northern
Norway Regional Health Authority Helse Nord
(number 2019/995 to MM). The funders had no
role in study design, data collection and analysis,
decision to publish, or preparation of the
manuscript.

**Competing interests:** The authors have declared
that no competing interests exist.

## Introduction

Potentially toxic elements like arsenic (As), mercury (Hg) and lead (Pb) are also referred to as
heavy metals. These metals are known for their adverse health effects to humans even at very
low concentrations [1–7]. The toxicity of these metals is considered to be high, hence catego-
rised as priority metals of public health relevance [8]. On the other hand, certain trace ele-
ments are needed by the body to sustain life hence required in small amounts in the daily diet
of animals including humans for proper growth and development [9]. However, imbalances in
trace elements, either deficiency or excess, have also been linked to adverse health effects dur-
ing pregnancy. For instance, very low concentrations of zinc (Zn) and very high concentra-
tions of copper (Cu) in the maternal blood is suggested to be linked to increased risk of
gestational diabetes mellitus (GDM) [10, 11]. Furthermore, very high concentration of Cu and
Zn has also been linked to increased cases of preterm births [12].

Prenatal exposure to potentially toxic metals and its effects on pregnancy outcomes remains
a public health concern. Evidence is increasing that these substances can be transferred from a
pregnant woman to her placenta through the umbilical cord and from a mother to a neonate
through breastfeeding [13, 14]. In this regard, the growing foetus and newborn are at a greatest
risk of the toxic effects of these potentially toxic metals.

Most of the biomonitoring studies on the potentially toxic metals and their effects on vul-
nerable population like pregnant women are usually conducted in the global north. However,
a knowledge gap exists in the southern hemisphere, especially within the African settings. Spe-
cifically, there is lack of knowledge on the levels of potentially toxic metals as well as trace ele-
ments in vulnerable populations like pregnant women and the associations with reproductive
health outcomes. This is due to the reason that most resources in these regions are usually
directed towards the research and control of communicable diseases. Potentially toxic metals
and trace elements have been used in Malawi for a long time. However, studies on exposure to
such compounds by susceptible groups like pregnant women and its associations on reproduc-
tive health outcomes are limited. This paper reports on the Malawi study that aimed at evaluat-
ing levels of potentially toxic metals and trace elements in maternal whole blood among
pregnant women in their third trimester of their pregnancy. The study also assessed the associ-
ation between maternal blood concentrations of these elements and maternal sociodemo-
graphic characteristics, lifestyle choices, and dietary habits. Furthermore, the study assessed
the association between maternal blood concentrations of potentially toxic metals as well as
trace elements and immediate birth outcomes (birth weight, birth length, head circumference
and gestational age).

## Materials and methods

### Study design, population, sites and data collection

The study design has been described previously elsewhere [15, 16]. Fig 1 shows three study
locations namely Ndirande Health Centre, Chiradzulu, and Thyolo District Hospitals, located
in the respective districts of Blantyre, Chiradzulu, and Thyolo. Chiradzulu and Thyolo district
hospitals are situated in rural settings, whereas Ndirande health centre is located in an urban
setting. All health facilities were located in the southern part of Malawi.

Recruitment of the study participants took place from August 2020 to July 2021. A total of
771 pregnant women in their third trimester attending antenatal services or admitted at the
labour wards of the included health facilities were assessed for eligibility from which 605 were
recruited. The inclusion criteria included: pregnant women in their late stages of third trimes-
ter of pregnancy, 16 years of age and above, permanent resident in the study districts and

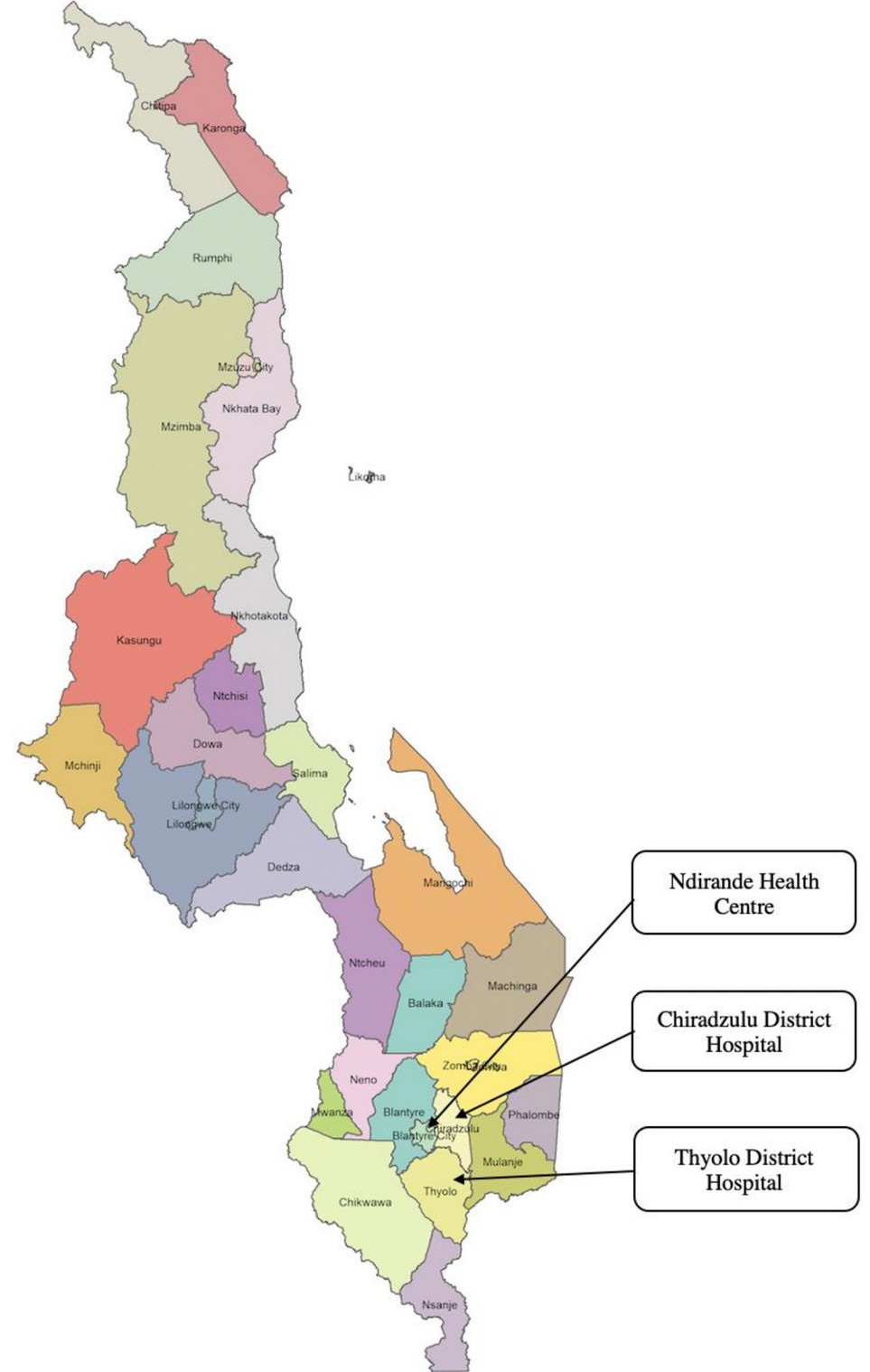

**Fig 1. Map of Malawi divided into districts and showing study sites (map generated in R; shapefile source: humdata.org).** https://data.humdata.org/dataset/cod-ab-mwi?; CC-BY 4.0 license [17].

willing to voluntarily sign an informed consent. Pregnant women with serious medical condition present at the time enrolment and with a high risk being referred to a tertiary health care level were excluded. Furthermore, pregnant women who were participating in other studies were also excluded from enrolment. Although a total of 771 pregnant women were assessed for eligibility, 605 recruited. This represented 308, 150 and 147 study participants recruited from Ndirande health centre, Thyolo district hospital and Chiradzulu district hospital respectively. A total of 506 pregnant women ultimately provided their blood samples and thus included in the final statistical analysis. Fig 2 shows the process of recruiting study participants starting from the assessment for eligibility to the final inclusion in the laboratory analysis.

Maternal socioeconomic characteristics, lifestyle, and newborn birth outcomes information were collected through questionnaire that was administered by a trained research nurse. The questionnaire also contained a food-frequency assessment section, with about 21 diet items ranging from meats, fish, vegetables and grains. The validity and reliability of the questionnaire used in the present study has been demonstrated in previous studies [18–20].

## Birth outcomes

Neonatal birth weight, body length, and head circumference were collected by the research nurses just after delivery. In this regard, a basin neonatal weighing scale was used for measuring the birth weight and the measuring tape for birth length and head circumference of the newborn. Gestational age was retrieved from the medical records (health passport).

## Maternal whole blood sample collection and preliminary analysis

Part of the procedure for the whole blood sample collection and preliminary analysis have been described in previous published articles from Malawi study on environmental and dietary exposure to persistent toxic substances (PTSs) in pregnancy and birth outcomes [15, 16]. In brief, maternal non-fasting whole blood samples were collected at an ideal period just before or after delivery (about 36 hours before or 12 hours after delivery). Venous whole blood was collected from the mother using a 6 mL dark blue top BD Vacutainers sample collection tube (BD REF # 368381). The samples were later transferred and aliquoted to three 2 mL Eppendorf polypropylene vials using disposable plastic pipettes. All collected biological samples were kept at a temperature between—35˚C to—20˚ C before shipment to University Hospital of North Norway (UNN), Division of Diagnostic Services, for further analysis.

## Analytical methods

The elemental analyses were conducted at the Environmental Pollutant Laboratory, Department of Laboratory Medicine, University Hospital of North Norway, using the dilute-and-shoot inductively coupled plasma mass spectrometry (ICP-MS) technique as described previously [21]. A robotic manipulator arm for transporting microtiterplates, an automated liquid handler (Tecan Freedom Evo 200, Männedorf, Switzerland) with an 8-channel liquid handler arm for conductive disposable tips, and a shaker (Bioshake, Quantifol Instruments GmBH, Jena, Germany) were used to dilute the samples. Dilution of 200 μL whole blood was done with a solution consisting of Milli-Q water (Millipore/Merck KGaA, Darmstadt, Germany), 10% v/v ammonia (Honeywell Fluka, Bucharest, Romania) and 2-propanol (Honeywell Fluka, Bucharest, Romania), 0.08% v/v Triton X-100 (Sigma/ Merck KGaA, Darmstadt, Germany) and 0.25 μg/L gold (Au; Inorganic Ventures, Christiansburg, VA, USA) followed by mixing on the shaker. The NexION 300D ICP-MS system (Perkin Elmer, Waltham, Massachusetts, USA) outfitted with an ESI-Fast SC2DX autosampler was utilised for the instrumental analysis.

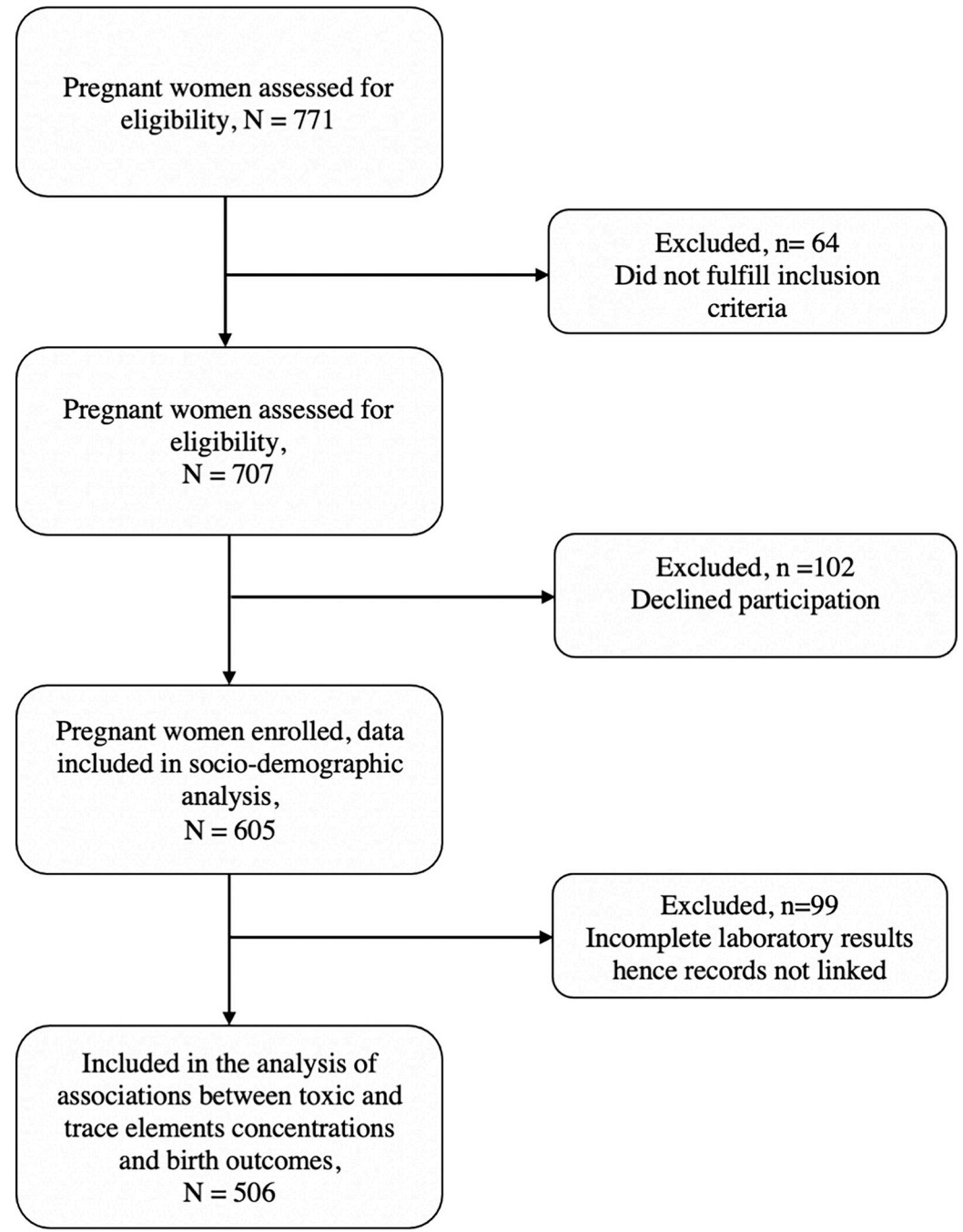

**Fig 2. Flow chart of study participants from eligibility to sample analyses.**

Through a T-piece an internal standard solution containing 20 μg/L rhodium and rhenium (Rhand Re Inorganic Ventures, Christiansburg, VA, USA) was added to the nebulizer.

The kinetic-energy-discrimination mode with helium as reaction gas was applied for instrumental analysis and measurements were conducted in triplicates. A matrix matched calibration curve with ClinCal whole blood calibration material from Recipe (Recipe, Munich, Germany) was used for quantitative determination of the elements in the samples. Elements

not present in the ClinCal material were spiked during sample preparation. The samples were analysed batchwise with each batch containing 28 samples, three ClinCal calibration samples diluted 1:100, 1:40 and 1:20 respectively, one calibration blank sample, four blank samples and two sets of ClinChek control material level 1, level 2 and level 3 from Recipe (Recipe, Munich, Germany), and Seronorm level 1, level 2 and level 3 (Sero, Billingstad, Norway) for quality assurance and quality control. Diluent blanks were used to control the background noise and instrumental carryover. Correlation coefficients of variations (CVs) for the different control samples were $\leq$ 7% (As), $\leq$5% (Cu), $\leq$ 10% (Hg), $\leq$ 6% (Ni), $\leq$ 3% (Pb), $\leq$ 6% (Se) and $\leq$ 5% (Zn). The lab successfully participates in the global quality control programme Quebec Multi-element External Quality Assessment Scheme (QMEQAS, Centre de Toxicologie du Quebec in Quebec, Canada) which involves elemental analysis in human whole blood.

## Statistical analysis

Following potentially toxic metals and trace elements were included in the statistical analysis: As, Cu, Pb, Hg, Ni, Se and Zn which showed a detection rate of $\geq$99%. Descriptive statistics were used to summarise sociodemographic data. We used Mann–Whitney U for continuous variables and Chi-square tests for categorial variables to compare the maternal and neonatal characteristics between urban and rural study participants. All potentially toxic metals and trace elements concentrations were transformed to natural logs for normalization prior to analysis.

Covariates, including maternal age, parity, drinking water source and area of residence, were selected based on a priori knowledge and existing research findings [20, 22]. Univariable linear regression analysis method used to determine additional maternal characteristics to be considered as covariates in the multivariable linear regression analysis. The maternal characteristics that were considered based on univariate analyses include maternal dietary habits and level of educational. Multivariable multiple linear regression analyses were performed to assess the associations between maternal and neonates´ characteristics with the concentrations of potentially toxic metal and trace elements. Furthermore, multivariable linear regression analyses were executed to assess the association between maternal blood levels of potentially toxic metals as well as trace elements and birth outcomes namely birth weight, birth length, head circumference and gestational age.

Multivariable linear regression models for potentially toxic and trace elements were adjusted for several covariates. In this regard, Se Cu and Pb multivariable linear regression models were adjusted for maternal age (in years), parity (nulliparous vs. primiparous or multiparous), maternal education level (no formal education/primary vs. secondary/tertiary), area of residence (urban vs. rural) and source of drinking water (tap water vs. lake/shallow well and borehole). Multivariable linear regression model for Zn was also adjusted for chicken eggs consumption frequency (< twice a week vs. $\geq$ twice a week) was addition to the list of the above listed covariates. Ni multivariable linear regression model was adjusted for maternal age (in years), parity (nulliparous vs. primiparous or multiparous), maternal education level (no formal education/primary vs. secondary/tertiary), area of residence (urban vs. rural), source of drinking water (tap water vs. lake/shallow well and borehole), beef and chicken eggs consumption frequencies (< twice a week vs. $\geq$ twice a week). Arsenic multivariable linear regression model was adjusted for maternal age (in years), parity (nulliparous vs. primiparous or multiparous), maternal education level (no formal education/primary vs. secondary/tertiary), area of residence (urban vs. rural), source of drinking water (tap water vs. lake/shallow well and borehole), and fresh fish consumption frequency (< twice a week vs. $\geq$ twice a week) and goat meat consumption frequency (< twice a week vs. $\geq$ twice a week). Mercury multivariable

linear regression model was adjusted for maternal age (in years), parity (nulliparous vs. primiparous or multiparous), maternal education level (no formal education/primary vs. secondary/tertiary), area of residence (urban vs. rural), source of drinking water (tap water vs. lake/ shallow well and borehole), fresh and dry fish consumption frequency ($<$ twice a week vs. $\geq$ twice a week) and green vegetables consumption frequency ($<$ twice a week vs. $\geq$ twice a week). Smoking status was not added as a covariate with a reason that only 2 out of 605 pregnant women recruited in this study reported that they were smokers, representing 0.33% only.

Statistical analyses were performed using Stata software for Mac (SE standard version 17; College Station, Texas, USA). A two-sided statistical significance was set at $p < 0.05$.

To assess the combined effects of exposure to potentially toxic metals and essential trace elements on birth outcomes, we employed Weighted Quantile Sum Regression (WQSR) analysis, a statistical method designed to analyze the combined effect of multiple correlated variables, often environmental exposures, while considering their potential joint contribution to a health outcome. We hypothesized that both toxic metals and essential trace elements would influence birth outcomes. Specifically, WQSR was used to explore the combined effects of As, Hg, and Pb concentrations in maternal blood on birth weight, birth length and gestational age at birth. A similar analysis was conducted to assess the combined impact of Ni, Se, Cu, and Zn on these birth outcomes.

### Ethics statement

The study was conducted in accordance with the Declaration of Helsinki and approved by the College of Medicine Research and Ethics Committee (COMREC)—Malawi (Ref #: P.11/18/ 2546) and Regional Committee for Medical and Health Research Ethics (REK)–Norway (Ref #: 355656 2020). The heads (District Health Officers) of the corresponding district health offices (Blantyre, Chiradzulu and Thyolo) in Malawi gave their consent for the study to be carried out in the chosen locations. Additionally, participation in the study was voluntary and was dependent on the mother's written informed consent form signed (or thumbprint witnessed by an independent individual for the illiterate) by the mother. All study participants were assigned study numbers for pseudo-anonymisation, which helped to guarantee confidentiality. The researcher ensured that the procedure of providing healthcare services at the health facility was not hampered by the study´s procedures.

**Inclusivity in global research.**   Additional information regarding the ethical, cultural, and scientific considerations specific to inclusivity in global research is included in the Supporting Information (S1 Checklist).

## Results

### Maternal socio-demographic and neonates´ data

Maternal and neonatal characteristics have been described in detail in our two previous published articles from the Malawi study [15, 16]. Maternal and neonatal characteristics by place of residence are shown in Table 1.

Briefly, the mean age of pregnant women recruited from the urban areas in the study was about 2 years higher as compared to their counterparts from the rural settings ($p < 0.001$). Over 80% of women underwent spontaneous vaginal delivery (SVD) in both settings. Just more than half of women were multiparas in the urban setting versus 34% in the rural area. The majority of women recruited from the urban area (61.4%) had attained either secondary or tertiary education in comparison to only 30.3% from rural areas. The majority of women in the present study were married in both settings.

Piped water was the major source of drinking water for urban residents (96.8%), compared to only 8.2% for the rural study participants. Consequently, the majority (91.8%) of women in

**Table 1. Characteristics of pregnant women and neonates.**

| Variable | N (Missing data) | Characteristic | Total | Place of residence | | p-value |
|---|---|---|---|---|---|---|
| | | | | Urban | Rural | |
| Total participants | | | 605 | 308 | 297 | |
| Age (Years) | 605 (0) | Mean (SD) | 24.8 (6.2) | 25.6 (5.7) | 23.9 (6.7) | <0.001 [a] |
| Mode of delivery (%) | 568 (37) | CS | 91 (16.0) | 53 (18.6) | 38 (13.4) | 0.111 [b] |
| | | SVD | 474 (83.5) | 229 (80.4) | 245(86.6) | |
| | | Breech | 2 (0.4) | 2 (0.7) | 0 (0.0) | |
| | | Vac extr | 1 (0.2) | 1 (0.4) | 0 (0.0) | |
| Parity (%) | 600 (5) | Nulliparous | 207 (34.5) | 61 (20.2) | 146 (49.2) | <0.001 [b] |
| | | Primiparous | 137 (22.8) | 86 (28.5) | 51 (17.2) | |
| | | Multiparous | 256 (42.7) | 155 (51.3) | 100 (33.6) | |
| Maternal education (%) | 604 (1) | None / primary | 325 (53.9) | 118 (38.6) | 207 (69.7) | <0.001 [b] |
| | | Secondary/ Tertiary | 278 (46.1) | 188 (61.4) | 90 (30.3) | |
| Marital Status (%) | 605 (0) | Married | 544 (90.0) | 283 (91.9) | 260 (87.6) | 0.081 [b] |
| | | Single | 61 (10.2) | 25 (8.1) | 37 (12.4) | |
| Previous breast feeding | 601 (4) | Yes | 355 (59.1) | 208 (68.0) | 148 (49.8) | <0.001 [b] |
| | | No | 246 (40.9) | 98 (32.0) | 148 (50.2) | |
| Drinking water source (%) | 602 (3) | Tap | 322 (53.5) | 298 (96.8) | 24 (8.2) | <0.001 [b] |
| | | Lake/ Shallow well | 137 (22.8) | 2 (0.7) | 135 (45.9) | |
| | | Borehole | 143 (23.8) | 8 (2.6) | 135 (45.9) | |
| Gestational age (weeks) | 541 (64) | Mean (SD) | 37.59 (1.53) | 37.47(1.43) | 37.71(1.62) | 0.075 [a] |
| Birth weight (Kg) | 571 (34) | Mean (SD) | 3.09 (0.46) | 3.18 (0.45) | 3.00 (0.46) | <0.001[a] |
| Birth length (cm) | 540 (65) | Mean (SD) | 45.19 (3.46) | 45.66(4.08) | 44.51(2.66) | 0.002 [a] |
| Head circumference (cm) | 540 (65) | Mean (SD) | 33.17 (1.83) | 33.14(1.94) | 33.19(1.72) | 0.845[a] |

[a] Mann–Whitney U test (urban vs rural).

[b] Chi-square test (urban vs rural).

Abbreviations: SD = standard deviation, CS = caesarian section, SVD = spontaneous vaginal delivery.

the rural setting relied on ground water, thus either shallow wells or boreholes as their drinking water source. Neonates from urban setting were on average 0.18 kg ($p < 0.001$) heavier and 1.2 cm longer ($p = 0.002$) compared to babies in rural areas. The mean head circumference for babies born from the two settings were similar, namely 33 cm in both settings.

**Maternal blood concentrations of potentially toxic metals and trace elements.** Arithmetic means (SD), geometric means (95% CI) and medians (max -min) for all potentially toxic metals and trace elements were computed (Table 2).

**Table 2. Maternal whole blood concentrations of potentially toxic metals and trace elements concentrations (µg/L wet weight; * mg/L wet weight).**

| Potentially toxic metals and trace elements | % > Detection Frequency | Method Detection Limits (MDLs) | Arithmetic Mean (SD) | Median (Min-Max) |
|---|---|---|---|---|
| Nickel | 100 | 0.147 | 0.91 (0.48) | 0.85 (0.40 to 7.66) |
| Selenium | 100 | 1.263 | 89.5 (29.2) | 87.8 (26.2 to 393) |
| Copper | 100 | 0.003* | 1641 (319) | 1649 (651 to 2980) |
| Zinc | 100 | 0.004* | 5751 (1341) | 5664 (2247 to 11240) |
| Arsenic | 99.3 | 0.060 | 0.41 (0.33) | 0.30 (0.03 to 2.15) |
| Mercury | 100 | 0.014 | 0.29 (0.19) | 0.24 (0.02 to 1.42) |
| Lead | 100 | 0.035 | 17.8 (18.5) | 12.9 (3.10 to 190) |

Abbreviations: SD = Standard Deviation; CI = Confidence Interval

Multivariable linear regression analysis results revealed that maternal age and area of residence were the main predictors of the concentrations of potentially toxic metals and trace elements in maternal whole blood. Table 3 shows the detailed assessment of the concentration of toxic and trace elements in maternal blood in relation to maternal characteristics.

Maternal age was positively associated with the concentrations of Cu ($\beta$ = 0.005; 95% CI: 0.001 to 0.009; $p$ = 0.023), As ($\beta$ = 0.014; 95% CI: 0.002 to 0.026; $p$ = 0.034) and Hg ($\beta$ = 0.015; 95% CI: 0.003 to 0.027; $p$ = 0.013). Area of residence was another main determinant of the concentrations of maternal potentially toxic metals and trace elements. On this, residing in rural

**Table 3. Multivariable linear regression of potentially toxic metals and trace elements in whole blood and maternal characteristics.**

| Maternal Characteristics | [a] Potentially toxic metals and trace elements | | | | | | |
|---|---|---|---|---|---|---|---|
| | Nickel | Selenium | Copper | Zinc | Arsenic | Mercury | Lead |
| | (n = 481) | (n = 490) | (n = 490) | (n = 484) | (n = 485) | (n = 473) | (n = 490) |
| **Maternal age** (years) | 0.004 | 0.004 | 0.005* | -0.0004 | 0.014* | 0.015* | 0.007 |
| **Parity** [b] | | | | | | | |
| Para 1 | -0.024 | -0.025 | -0.034 | 0.030 | 0.037 | 0.166* | -0.07 |
| Multiparity | -0.054 | -0.011 | -0.035 | 0.005 | 0.010 | -0.020 | -0.108 |
| **Education of mothers** [c] | -0.005 | 0.026 | -0.002 | 0.002 | 0.013 | 0.024 | -0.030 |
| **Area of residence** [d] | 0.140* | 0.003 | -0.001 | 0.054 | 0.498** | -0.223 | -0.091 |
| **Source of drinking water** [e] | | | | | | | |
| Lake/ shallow well | -0.079 | -0.071 | -0.039 | -0.006 | 0.143 | -0.010 | 0.223* |
| Borehole | -0.027 | -0.062 | 0.028 | -0.014 | -0.166 | 0.170 | 0.119 |
| **Beef Consumption** [f] | -0.006 | - | - | | -0.063 | - | - |
| **Chicken eggs consumption** [g] | -0.035 | - | - | 0.078* | - | - | - |
| **Fresh fish Consumption** [h] | -0.026 | - | - | - | 0.100** | 0.183 | - |
| **Dry fish Consumption** [i] | - | - | - | - | - | 0.172 | - |
| **Goat Consumption** [j] | - | - | - | - | -0.108 | | - |
| **Green Veg Consumption** [k] | - | - | - | - | - | -0.720* | - |

Values shown are coefficients of multiple regression analyses coefficient.

[a] Potentially toxic metal and trace element concentrations were transformed to natural logs before analysis.

[b] Para 0 as a reference category.

[c] None and primary educational level as reference category.

[d] Urban as reference category. [e] Tap water as reference category.

[f,g,h,ai j & k] Less than twice a week as reference categories. Ni model was adjusted for maternal age, education, area of residence, source of drinking water and beef/chicken eggs/ fresh fish diet. Selenium, Copper and Lead models were adjusted for maternal age, parity, maternal education level, area of residence and source of drinking water. Zinc model was adjusted for maternal age, parity, maternal education level, area of residence and source of drinking water and egg diet. Arsenic model was adjusted for maternal age, parity, maternal education level, area of residence and source of drinking water and fresh fish/beef/goat meat diet. Mercury model was adjusted for maternal age, parity, maternal education level, area of residence and source of drinking water and fresh and dry fish and green vegetables diet.

*$p < 0.05$

** $p < 0.001$.

areas was significantly associated with high concentrations of Ni in maternal whole blood ($\beta$ = 0.140; 95% CI: 0.019 to 0.262; $p$ = 0.024). Similarly, rural residency was also associated with high maternal blood As concentrations ($\beta$ = 0.498; 95% CI: 0.236 to 0.759; $p$ <0.001).

Primiparity was associated with high concentrations of Hg ($\beta$ = 0.166; 95% CI: 0.013 to 0.320; $p$ = 0.034). However, no associations were observed with multiparity. In reference to tap water, the use of water from lake or shallow well was statistically significant associated with increased concentrations of Pb ($\beta$ = 0.223; 95% CI: 0.023 to 0.359; $p$ = 0.032). Increased consumption of chicken eggs was positively associated with high maternal blood Zn concentrations ($\beta$ = 0.078; 95% CI: 0.028 to 0.127; $p$ = 0.002). Positive statistically significant association was observed between maternal blood As concentrations and fresh fish consumption ($\beta$ = 0.200; 95% CI: 0.081 to 0.318; $p$ = 0.001). Variance inflation factors (VIF) computed in multicollinearity testing for all models were below 10, indicating moderate multicollinearity between the independent variables hence acceptable [23]. No associations were observed between Se with each of the maternal characteristics that were assessed in the present study.

## Concentrations of potentially toxic metals and trace elements in maternal blood versus birth outcomes

Multivariable linear regression analysis indicated high maternal blood Ni concentrations were associated with increased birth weight ($\beta$ = 0.063; 95% CI: 0.003 to 0.124; $p$ = 0.047), birth length ($\beta$ = 0.009; 95% CI: 0.001 to 0.018 $p$ = 0.026), head circumference ($\beta$ = 0.018; 95% CI: 0.002 to 0.034; $p$ = 0.029) and gestational age ($\beta$ = 0.023; 95% CI: 0.002 to 0.044; $p$ = 0.035). Similarly, WQSR analysis revealed a positive association between essential elements and neonatal birth weight with effect size of 0.018 (0.0042 to 0 032; $p$ = 0 010).

Results from multivariable linear regression analysis showed that maternal whole blood As concentrations were negatively association with neonatal birth length ($\beta$ = -0.018; 95% CI: -0.035 to -0.0001; $p$ = 0.048) and head circumferences ($\beta$ = -0.067; 95% CI: - 0.098 to -0.030; $p$ <0.001). Similarly, Pb concentrations were also associated with decrease in head circumference ($\beta$ = -0.048; 95% CI: -0.079 to -0.017; $p$ = 0.002) and birth length ($\beta$ = -0.020; 95% CI: -0.036 to -0.004; $p$ = 0.016). No statistically significant associations were observed for Se, Cu, Zn and Hg with birth outcomes. However, WQSR analysis did not show any statistically significant associations between potentially toxic metals and all birth outcomes. The associations for maternal blood concentrations of potentially toxic metals and trace element concentrations in relation to birth outcomes are shown in Fig 3.

## Ad-hoc analyses on maternal blood concentrations of potentially toxic metals and trace elements versus maternal characteristics

Ad-hoc analyses were also conducted by stratifying pregnant mothers according to the gender of their neonates. On this, statistically significant positive association was observed between the maternal age and Cu blood concentrations ($\beta$ = 0.007; 95% CI: 0.0004 to 0.014; $p$ = 0.038 for mother to male neonates only. Living in rural areas was significantly associated with high concentrations of Ni in maternal whole blood for pregnant women who gave birth to both female ($\beta$ = 0.461; 95% CI: 0.065 to 0.856; $p$ = 0.023) and male neonates ($\beta$ = 0.525; 95% CI: 0.155 to 0.894; $p$ = 0.006). Table 4 shows the detailed assessment of the concentration of potentially toxic metals and trace elements in maternal blood in relation to maternal characteristics stratified by the gender of the neonates.

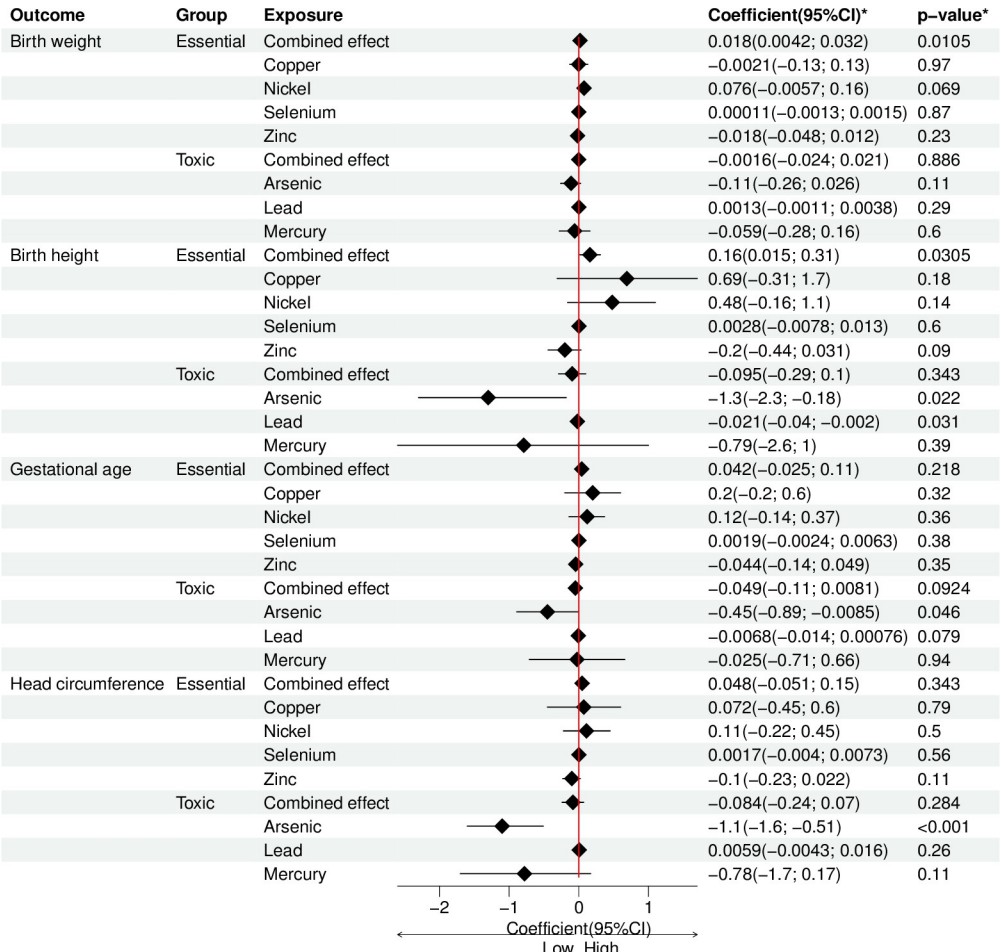

**Fig 3. The change in birth outcomes in relation to the changes in potentially toxic metal and trace element concentrations.** For multivariable linear regression analysis, concentrations of potentially toxic metals and trace element concentrations were transformed to natural logs prior to analysis. Multiple linear regression coefficients are displayed with 95% confidence interval. Ni model was adjusted for maternal age, parity, maternal education level, area of residence, source of drinking water and beef/chicken eggs diet. Se Cu and Pb models were adjusted for maternal age, parity, maternal education level, area of residence and source of drinking water. Zn model was adjusted for maternal age, parity, maternal education level, area of residence and source of drinking water and chicken eggs diet. As model was adjusted for maternal age, parity, maternal education level, area of residence and source of drinking water and fresh fish/goat meat diet. Hg model was adjusted for maternal age, parity, maternal education level, area of residence and source of drinking water, fresh and dry fish and green vegetables diet. Abbreviations: Ni = nickel; Se = selenium; Cu = copper; Zn = zinc; As = arsenic; Hg = mercury; Pb = lead. For WQSR analysis, combined effect models of maternal blood concentrations of potentially toxic metals or essential elements and birth outcomes was adjusted for age, source of drinking water, maternal, educational level and the intake of fresh fish and goat meat.

## Ad-hoc analyses on the concentrations of potentially toxic metals and trace elements in maternal whole blood versus birth outcomes

When the models were repeated after stratification by gender of the neonates, positive statistically significant associations were observed between maternal blood Ni concentrations and birth weight ($\beta$ = 0.10; 95% CI: 0.01 to 0.19; $p$ = 0.024), birth length ($\beta$ = 0 .012; 95% CI: 0.01 to 0.03 $p$ = 0.004) and gestational age ($\beta$ = 0.03; 95% CI: 0.005 to 0.064; $p$ = 0.023) for male neonates only. Negative significant association was observed between maternal blood As concentration and head circumference for male ($\beta$ = -0.09; 95% CI: -0.14 to -0.04; $p$ = 0.001) but not female neonates. In contrast, maternal blood Pb concentrations were associated with decrease

**Table 4. Multivariable linear regression of potentially toxic metal and trace element concentrations in whole blood and maternal characteristics stratified by neonate gender.**

| Maternal characteristics | | [a] Potentially toxic metals and trace elements | | | | | | | |
|---|---|---|---|---|---|---|---|---|---|
| | | Nickel | Selenium | Copper | Zinc | Arsenic | Mercury | Lead | |
| | | (n = 244) | (n = 249) | (n = 249) | (n = 246) | (n = 246) | (n = 241) | (n = 249) | |
| **Maternal age (years)** | | -0.001 | 0.005 | 0.004 | -0.004 | 0.013 | 0.014 | 0.006 | **Pregnant mothers carrying female foetuses** |
| **Parity [b]** | | | | | | | | | |
| | Para 1 | 0.049 | -0.084 | -0.024 | 0.088 | 0.081 | 0.18 | -0.05 | |
| | Multiparity | 0.066 | -0.076 | -0.028 | 0.062 | 0.088 | 0.036 | -0.122 | |
| **Education of mothers [c]** | | 0.022 | -0.032 | 0.035 | 0.013 | -0.035 | -0.101 | -0.077 | |
| **Area of residence [d] (Urban vs Rural)** | | 0.242* | 0.03 | -0.016 | 0.076 | 0.461* | -0.096 | -0.066 | |
| **Source of drinking water [e]** | | | | | | | | | |
| | Lake/ shallow well | -0.151 | -0.06 | -0.043 | -0.036 | 0.151 | -0.211 | 0.303 | |
| | Borehole | -0.029 | -0.09 | 0.065 | 0.011 | 0.064 | 0.077 | 0.248 | |
| **Beef consumption frequency [f]** | | 0.02 | - | - | | -0.07 | - | - | |
| **Chicken eggs consumption frequency [g]** | | 0.014 | - | - | 0.107* | - | - | - | |
| **Fresh fish consumption frequency [i]** | | 0.033 | - | - | - | 0.236* | 0.235* | - | |
| **Dry fish consumption frequency [j]** | | - | - | - | - | - | 0.202 | - | |
| **Goat consumption frequency [k]** | | - | - | - | - | -0.091 | - | - | |
| **Green Veg consumption frequency [l]** | | - | - | - | - | - | 0.284 | - | |
| | | (n = 215) | (n = 219) | (n = 219) | (n = 216) | (n = 217) | (n = 210) | (n = 219) | |
| **Maternal age (years)** | | 0.009 | 0.001 | 0.007* | 0.02 | 0.011 | 0.012 | 0.01 | **Pregnant mothers carrying male foetuses** |
| **Parity [b]** | | | | | | | | | |
| | Para 1 | -0.09 | 0.081 | -0.048 | -0.008 | 0.078 | 0.238 | -0.132 | |
| | Multiparity | -0.167 | 0.093 | -0.068 | -0.039 | 0.022 | 0.017 | -0.174 | |
| **Education of mothers [c]** | | -0.043 | 0.083 | -0.038 | 0.006 | 0.049 | 0.159 | 0.013 | |
| **Area of residence [d] (Urban vs Rural)** | | 0.05 | -0.003 | -0.011 | 0.022 | 0.525* | -0.301 | -0.155 | |
| **Source of drinking water [e]** | | | | | | | | | |
| | Lake/ shallow well | -0.027 | 0.038 | -0.043 | 0.031 | 0.144 | 0.141 | 0.261 | |
| | Borehole | -0.038 | -0.041 | 0.009 | -0.026 | -0.374* | 0.202 | 0.021 | |
| **Beef consumption frequency [f]** | | -0.032 | - | - | | -0.078 | - | - | |
| **Chicken eggs consumption frequency [g]** | | -0.06 | - | - | 0.063 | - | - | - | |
| **Fresh fish consumption frequency [i]** | | 0.088* | - | - | - | 0.164 | 0.128 | - | |
| **Dry fish consumption frequency [j]** | | - | - | - | - | - | 0.068 | - | |
| **Goat consumption frequency [k]** | | - | - | - | - | -0.111 | - | - | |
| **Green Veg consumption frequency [l]** | | - | - | - | - | - | -1.078** | - | |

Values shown are coefficients of multiple regression analyses.

[a] Potentially toxic metal and trace element concentrations were transformed to natural logs before analysis.

[b] Para 0 as a reference category.

[c] None and primary educational level as reference category.

[d] Urban as reference category.

[e] Tap water as reference category.

[f,g,h,i j & k] Less than twice a week as reference categories. Ni model was adjusted for maternal age, education, area of residence, source of drinking water and beef/chicken eggs/ fresh fish diet. Se, Cu and Pb models were adjusted for maternal age, parity, maternal education level, area of residence and source of drinking water. Zn model was adjusted for maternal age, parity, maternal education level, area of residence and source of drinking water and egg diet. Arsenic model was adjusted for maternal age, parity, maternal education level, area of residence and source of drinking water and fresh fish/beef/goat meat diet. Hg model was adjusted for maternal age, parity, maternal education level, area of residence and source of drinking water and fresh and dry fish and green vegetables diet.

*$p < 0.05$

** $p < 0.001$.

in head circumference ($\beta$ = -0.059; 95% CI: -0.11 to -0.01; $p$ = 0.018) and birth length ($\beta$ = -0.029; 95% CI: -0.055 to -0.004; $p$ = 0.022) for female neonates only.

## Discussion

Area of residence and maternal age emerged as the primary determinants of the levels of potentially toxic metals and trace elements in maternal blood. Notably, increased concentrations of Ni were associated with positive birth outcomes as compared to other potentially toxic metals. In this regard, higher maternal blood Ni concentrations were associated with increased neonatal birth weight, birth length and head circumference. In contrast, increased concentrations of As and Pb were associated with lower birth length and smaller head circumference.

### Potentially toxic metal and trace element concentrations and maternal characteristics

We observed a significant increase in blood Cu, As and Hg concentrations with increasing maternal age. This could be attributed to the advanced age as older women are more likely to have more exposure to these elements over the years compared to young women. The positive association between Cu and maternal age align with results from the study by Xu et al [20].

Notably, increased maternal blood Zn concentrations were associated with high maternal egg consumption. This could be because of the fact that chicken eggs are rich in a number of nutrients namely; calcium (Ca), iron (Fe), magnesium (Mg), phosphorus (P), Se, sodium (Na) and Zn [24, 25]. Living in rural areas was significantly associated with increased levels of maternal As. These results were rather unexpected as some studies have observed higher concentrations of As in urban and industrial areas as compared to rural areas. In contrast, Malawi is not a highly industrialized country as most of the products that are used in are imported hence the potential of contamination due to manufacturing industries is very low. However, these results from the Malawi study may be attributed to continuous use of fertilizers for farming in the rural setting. On this, Malawi´s economy is highly dependent on farming and the majority of the farms are situated in rural areas. A number of research studies have suggested the use of phosphate fertilizer as the main source of As contamination to the soils and drinking water sources [26–28]. Secondly, results from Malawi study suggests that differences in maternal blood As concentrations between urban and rural study participants may be attributed to different lifestyles in terms of drinking water sources and food as oral ingestion is the common human exposure to As [29, 30]. The present study investigated the total As concentrations, which is a mixture of different organic and inorganic As-species, in order to assess a first picture of the As load in maternal blood samples. The results of our Malawi study give an indication that As could be a problem, and especially in the rural setting. For deeper investigation of the As-composition and further evaluation regarding toxicity levels a speciation analysis is recommended as this was beyond the scope of the present work. However, there is also need for follow-up research to investigate and compare the concentrations of As in different foods products consumed in areas from which maternal blood samples were collected to ascertain this hypothesis. Although the concentrations of As observed in this study was lower as compared to most of the studies around the globe it was noted that the maternal blood concentrations were positively associated with fresh fish consumption. This result supports the scientific notion that As may also bioaccumulates in fresh water fish [31].

Similarly, higher concentrations of Ni in maternal blood were observed in the blood of rural as compared to urban pregnant women. These results could be attributed to the presence of a very big landfill in one of the rural sites that was included in the present study namely, Chiradzulu district. This hypothesis is supported by results from research studies conducted in

landfills in Cameroon [32], Nigeria [33, 34], Bangladesh [35] and India [36] that observed high concentrations of Ni in solid waste landfills.

## Potentially toxic metals and trace elements and birth outcomes

Published research studies on biological role of Ni in the human body and the association between maternal Ni exposure and birth outcomes is limited. Furthermore, in addition to venous blood, there is a diverse range of samples that may be collected for Ni analysis ranging from urinary, placenta and umbilical cord blood samples. The maternal blood Ni concentrations in the present study were positively associated with birth weight, birth length, head circumference and gestational age. Although the biological function of Ni is still unclear, it is speculated that this element is required for the body for important biological processes. Specifically, Ni is believed to take part in protein structure and function. Furthermore, Ni is suggested to be linked to the process of certain enzymes that are connected to the breakdown or use of glucose [37, 38]. In this regard, adequate concentrations of Ni in the maternal blood would enhance favorable birth outcomes. Although Ni has some benefits to humans as stated above, it is also important to note that very high lconcentrations may also be associated with adverse health effects. For instance, high exposures to Ni is believed to cause damage in the genetic information, defects in developing foetus, obstruct function of the immune system and cause some forms of cancer [39]. The results of the present study that suggested positive association between Ni and birth weight are consistent with a study conducted by Michael et al. in Israel [40]. However, Zheng et al. [41] and Odland et al. [42] in their separate studies conducted in China and Russia respectively, observed no significant association between prenatal Ni and adverse birth outcomes. It should be noted that maternal umbilical cord blood samples were analysed in the Chinese study, while urinary samples were used in the Russian study.

There are conflicting results in the literature regarding the association between maternal As concentrations and birth outcomes. Some studies have failed to show any association between these [43, 44]. On the other hand, some studies have suggested inverse associations between maternal As concentrations and birth outcomes [45–48]. Arsenic is found naturally in the earth's crust and widely distributed throughout the environment. This element is very toxic in its inorganic form (arsenite or arsenate) and diet and drinking water rank among the main sources of exposure to humans. In contrast, organic As such as such as arsenobetaine are relatively non-toxic and mostly found in seafoods [49]. In the present study, maternal blood As (organic and inorganic) concentrations were negatively associated with birth length and head circumference. Our results are similar to what studies conducted by Guan et al. and Mullin et al. found in Peru [50] and Mexico [45] respectively.

Similar to As, significant inverse associations were also observed between maternal Pb concentrations with neonates' birth length and head circumference. The arithmetic mean concentration for Pb detected in the present study was 17.8 $\mu$g/L. Although there is no known safe level of exposure to Pb according to the 2021 revised CDC Pb reference concentration [51] there is a need for continued biomonitoring surveys and more studies to find out the sources of both environmental and dietary Pb exposure as some traces of Pb were still detected in the maternal blood in study. The results in the present study on the associations between maternal blood As concentrations and some birth outcomes are consistent with the study by Xu et al. conducted in Salta, Argentina [20]. In the Argentinean study, maternal blood Pb concentrations were also inversely associated with birth length. Another similar result to the present study was found in the ALSPAC study by Taylor et al. Similarly, ALSPAC study where inverse association between maternal blood Pb concentration and head circumference was observed [52]. The above observed associations in the present study were anticipated as a number of

epidemiological studies have suggested the link between Pb exposure and a number of adverse pregnancy outcomes, namely low birth weight, preterm birth, small for gestational age [20, 53–56].

Zn is essential in a number of important biological processes in the human body [57]. However, we did not find any statistically significant associations between Zn exposure and any of the birth outcomes assessed in the present study. These results are consistent to studies conducted by Rahman et al. and Tamura et al. [58–60]. In contrast, Boskabadi et al., 2021 reported a weak positive association between maternal blood Zn concentrations and birth weight [61].

No associations were observed between maternal blood Se concentrations and birth weight, birth length, head circumference and gestational age. However, the results from the present study are in contrast to another previous published study form Eastern China that suggested positive association between maternal Se concentration with birth weight [62]. These discrepant research results may be explained by the fact that the geometric mean Se concentration in the present study were very low (85.3 $\mu$g/L) as compared to a study conducted in Eastern China (143 $\mu$g/L).

## Comparison with similar studies

Maternal blood concentrations of most potential toxic metals and trace elements examined in the Malawi study, except for Hg, were generally low as compared to a number of similar published studies worldwide. The details for the arithmetic mean, geometric mean or median concentrations for maternal blood potentially toxic metals and trace elements for the present study in comparison to other similar studies globally are given in Table 5.

Notably, maternal blood As and Hg concentrations were low as compared to published studies from Argentina, Brazil, French Guiana, Bolivia, Peru, Suriname, Spain, Norway, South Africa, Benin, Japan and Australia [20, 22, 63, 64, 67, 70–72, 76–78, 81, 82]. This was rather expected as Malawi is a landlocked country with no access to the sea. A number of studies have shown high levels of As and Hg in their populations have access to the sea with high consumption of sea food [83–86].

The WQSR analysis indicates that the relationships between the maternal concentrations of potentially toxic metals as well as essential trace elements and birth outcomes are complex. These findings underscore the importance of considering combined effects of multiple metal exposures on birth outcomes, as simpler linear models may fail to capture the true nature of these relationships.

## Study strengths and limitations

The sample size for the present study is large as compared to many similar studies conducted globally. Furthermore, samples were collected from both urban and rural settings and by trained professional healthcare personnel guaranteeing a reproducible sample collection throughout the study.

While our study possesses several strengths, it is important to recognize its limitations. Firstly, causality could not be ascertained as this was a cross-sectional study. Furthermore, self-reported diet data may have introduced reporting bias, affecting the association between maternal diet and the concentrations of toxic metals and trace elements. Moreover, despite comprehensive inclusion and exclusion criteria for the participants, we did not record data on the characteristics of those who declined to participate, which may potentially introduce selection bias. However, the proportion of the pregnant women that declined to take part in this study was very low (14.4%). Lastly, our study may lack statistical power to detect associations on subgroup analyses as this was not considered in the initial statistical power calculations.

**Table 5. Concentration of maternal potentially toxic metals and trace elements in the Malawi study compared to similar global studies (μg/L).**

| | Location of the Study | Sampling Period | Sample Size | Arsenic | Mercury | Lead | Copper | Nickel | Selenium | Zinc |
|---|---|---|---|---|---|---|---|---|---|---|
| **Africa** | Present Study [a] | 2020–2021 | 605 | 0.42 | 0.29 | 17.8 | 1640 | 0.91 | 89.5 | 5751 |
| | Present Study [b] | 2020–2021 | 605 | 0.32 | 0.24 | 14.2 | 1610 | 0.86 | 85.3 | 5591 |
| | Present Study [c] | 2020–2021 | 605 | 0.30 | 0.24 | 12.9 | 1649 | 0.85 | 87.8 | 5664 |
| | Benin [63] [b] | 2015 | 60 | - | - | 38 | 1544 | - | - | 5215 |
| | South Africa [64] [b] | 2005–2006 | 62 | 0.57 | 0.65 | 23 | 1730 | - | 104 | 6290 |
| **Asia** | Iran [65] [a] | 2016–2017 | 206 | - | - | 49.6 | - | 96.6 | - | - |
| | China [66] [b] | 2016 | 915 | 3.88 | - | 9.96 | | | 133 | |
| | Australia [67] [c] | 2008–2011 | 173 | 1.26 | - | - | 1252 | < 2.0 | 88.2 | 2330 |
| | Shanx, China [68] [c] | 2010 | 215 | 0.52 | 0.26 | 24.5 | - | - | - | - |
| | China [69] [c] | 2006–2007 | - | 3.81 | - | 64.3 | - | - | - | 6312 |
| | Eastern China [62] [b] | - | 209 | - | | 39.5 | - | - | 143 | - |
| | Japan [70] [c] | 2001–2006 | 649 | 4.06 | - | 10.8 | 1289 | - | 176 | 4769 |
| **Europe** | Spain [22] [c] | 2016–2017 | 40 | 1.8 | 1.8 | 12 | 1664 | - | 107 | 6708 |
| | Norway [71] [b] | 2007–2009 | 211 | 1.8 | 1 | 9.2 | 1780 | - | 72 | 5480 |
| | Bolivia [72] [b] | 2007–2008 | 419 | 6.14 | - | - | - | - | 114 | - |
| **North America** | Puerto Rico [73] [b] | 2011–2017 | 1183 | 0.34 | 1.2 | 3.3 | 1552 | 1 | - | 4682 |
| | USA [74] [b] | 2009–2011 | 211 | 0.45 | 0.45 | 8.9 | - | - | - | - |
| | Mexico [45] [b] | 2007–2008 | 299 | - | - | 23.8 | - | - | - | - |
| | Costa Rica [75] [b] | 2010–2011 | 418 | - | - | | - | - | - | - |
| **South America** | Suriname [76] [c] | 2016 | 76 | - | 3.88 | 47.3 | - | - | - | - |
| | French Guiana [77] [b] | 2013 | 531 | - | - | 32.6 | - | - | - | - |
| | Los Cobres, Argentina [78] [c] | 2012–2013 | 169 | 2.2 | - | 21 | - | - | 86 | 6100 |
| | Salta, Argentina (Smokers) [20] [b] | 2011–2012 | 498 | 0.55 | 0.62 | 15.8 | 1782 | - | 129.2 | 6682 |
| | (Non smokers) [b] | | - | - | 0.6 | 15 | 1766 | - | 128.8 | 6720 |
| | Ushuaia, Argentina (Smokers) [20] [b] | 2011–2012 | 198 | 0.63 | 0.35 | 10.1 | 1688 | - | 80.1 | 7633 |
| | (Non smokers) [b] | | - | - | 0.34 | 9.81 | 1781 | - | 80.6 | 7815 |
| | Canada [79] [c] | 2008–2011 | 1673 | - | 0.56 | 5.6 | - | - | - | - |
| | Colombia [80] [c] | 2009–2010 | 381 | - | - | 9.5 | - | - | - | - |
| | Brazil [81] [c] | 2007–2008 | 155 | 0.6 | 0.6 | 16.2 | 1735 | - | 64 | 6420 |
| | Peru [82] [c] | 2004–2005 | 204 | - | - | - | - | - | - | - |

[a] Concentration expressed as Arithmetic mean

[b] Concentrations expressed as Geometric mean

[c] Concentration expressed as Median

## Conclusions

This work presents an important information on the main determinants of maternal blood toxic metals and trace elements and its association with birth outcomes among pregnant women from southern Malawi. Maternal age, followed by area of residence were the main determinant of exposure to both toxic metals and trace elements. Although the concentrations of most toxic metals were low as compared to the global trend, some inverse associations were observed. In this regard, there is a need to monitor the concentrations and establish ways of averting exposure, especially among vulnerable populations including pregnant women. Furthermore, there is a need for biomonitoring surveys to establish reference concentrations for both potentially toxic metals and trace elements in this region and in Malawi in general.

## Supporting information

**S1 Checklist. Inclusivity in global research.**
(DOCX)

## Acknowledgments

Special thanks go to the Laboratory Managers; Lydia Moyo and Jones Kadewere for coordinating sample collection and managing laboratory technicians in Malawi health facilities. Many thanks to research nurses; nursing coordinator, Blessings Kadzuwa for managing study nurses; Eunice Makwinja, Matilda Tewesa and Chifundo Mpawa for obtaining consent and administering study questionnaires. Laboratory scientists; Christopher Khungwa, Yohane Chisale and Ollings Mughandira / Laboratory technicians; Judith Mponda, Dyson Pindeni, Agnala Epikaizo Jumbe, Jamester Chilunjika for assisting in sample collection, storage and preliminary analysis. Our gratitude goes to Cynthia Amin for entering and managing the data. Laboratory technicians at the University Hospital of North Norway; Merete Linchausen Skar, Arntraut Götsch and Christina Ripman Hansen for elements analysis.

## Author Contributions

**Conceptualization:** Mphatso Mwapasa, Jon Øyvind Odland.

**Data curation:** Mphatso Mwapasa.

**Formal analysis:** Mphatso Mwapasa, Shanshan Xu, Augustine Choko.

**Funding acquisition:** Jon Øyvind Odland.

**Investigation:** Mphatso Mwapasa, Bertha Magreta Chakhame.

**Methodology:** Mphatso Mwapasa, Sandra Huber.

**Project administration:** Alfred Maluwa, Jon Øyvind Odland.

**Supervision:** Alfred Maluwa, Halina Röllin, Jon Øyvind Odland.

**Validation:** Mphatso Mwapasa, Shanshan Xu.

**Visualization:** Shanshan Xu.

**Writing – original draft:** Mphatso Mwapasa.

**Writing – review & editing:** Mphatso Mwapasa, Shanshan Xu, Bertha Magreta Chakhame, Alfred Maluwa, Halina Röllin, Augustine Choko, Sandra Huber, Jon Øyvind Odland.

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
