## [Decision Letter · Decision Letter 0]

5 Aug 2024

PGPH-D-24-00888

Concentrations of potentially toxic and trace elements in pregnant women and association with birth outcomes: A cross-sectional study in Malawi

Dear Dr. Mwapasa,

Thank you for submitting your manuscript to PLOS Global Public Health. After careful consideration, we feel that it has merit but does not fully meet PLOS Global Public Health’s publication criteria as it currently stands. Therefore, we invite you to submit a revised version of the manuscript that addresses the points raised during the review process.

We look forward to receiving your revised manuscript.

Kind regards,

Gauri Desai, PhD

Academic Editor

Journal Requirements:

2. We have amended your Competing Interest statement to comply with journal style. We kindly ask that you double check the statement and let us know if anything is incorrect. 

3. We do not publish any copyright or trademark symbols that usually accompany proprietary names, eg (R), (C), or TM  (e.g. next to drug or reagent names). Please remove all instances of trademark/copyright symbols throughout the text, including ® on page 6.

4. In the online submission form, you indicated that "Data would be made available on request from the corresponding author.". 

a. In a public repository, 

b. Within the manuscript itself, or 

c. Uploaded as supplementary information.

5. Figure 1: please (a) provide a direct link to the base layer of the map (i.e., the country or region border shape) and ensure this is also included in the figure legend; and (b) provide a link to the terms of use / license information for the base layer image or shapefile. We cannot publish proprietary or copyrighted maps (e.g. Google Maps, Mapquest) and the terms of use for your map base layer must be compatible with our CC-BY 4.0 license. 

Additional Editor Comments (if provided):

Reviewers' comments:

Reviewer's Responses to Questions

**Comments to the Author**

1. Does this manuscript meet PLOS Global Public Health’s publication criteria? Is the manuscript technically sound, and do the data support the conclusions? The manuscript must describe methodologically and ethically rigorous research with conclusions that are appropriately drawn based on the data presented.

Reviewer #1: No

2. Has the statistical analysis been performed appropriately and rigorously?

Reviewer #1: No

3. Have the authors made all data underlying the findings in their manuscript fully available (please refer to the Data Availability Statement at the start of the manuscript PDF file)?

Reviewer #1: Yes

4. Is the manuscript presented in an intelligible fashion and written in standard English?

Reviewer #1: Yes

5. Review Comments to the Author

Reviewer #1: From my perspective, this article is notably interesting due to the type of population it studies, as it provides valuable information about a community that has been less explored in the scientific literature. It is particularly relevant to investigate how environmental exposures influence childhood growth and development in these less-studied areas. Delving deeper into this aspect could reveal specific patterns and effects that have not been identified in more widely investigated contexts.

However, I have identified several areas for improvement that could enhance the rigor and validity of the study. Firstly, there is a notable lack of review and application of the most current methodologies in environmental epidemiology. Advanced techniques such as Bayesian Kernel Machine Regression (BKMR), Weighted Quantile Sum Regression (WQSR), and gComputation are powerful tools for analyzing the potential effects of heavy metal exposure and its relationship with child health outcomes. These methodologies allow for a more robust and precise evaluation of multiple exposures and their interactive effects, significantly improving the quality of the analysis.

Additionally, in the sample analysis methodology section, the Limit of Detection (LOD) for each analyzed metal is not specified. The omission of this information is critical, as the LOD determines the sensitivity of the analysis and, consequently, the accuracy with which low concentrations of heavy metals can be detected. It is essential to report the percentage of samples below the LOD and to describe in detail the treatment of these data, as strategies for handling values below the LOD can significantly influence the results and their interpretation.

Furthermore, I have noticed that the discussion is organized into sections, which is not typical for studies of this type. Although a sectioned structure can offer greater clarity in presenting results and conclusions, it can also fragment the discussion and hinder the integration of findings into a coherent narrative. I believe that a more fluid and continuous discussion could facilitate the understanding and impact of the presented results.

Incorporating these elements would not only strengthen the scientific validity of the study but also allow for greater comparability with similar studies, thus facilitating the integration of the findings into the broader body of knowledge on environmental epidemiology and child health.

6. PLOS authors have the option to publish the peer review history of their article (what does this mean?). If published, this will include your full peer review and any attached files.

**Do you want your identity to be public for this peer review?** For information about this choice, including consent withdrawal, please see our Privacy Policy.

Reviewer #1: No

---

## [Editor Report · Decision Letter 1]

2 Oct 2024

Concentrations of potentially toxic metals and trace elements in pregnant women and association with birth outcomes: A cross-sectional study in Malawi

PGPH-D-24-00888R1

Dear Dr Mwapasa,

We are pleased to inform you that your manuscript 'Concentrations of potentially toxic metals and trace elements in pregnant women and association with birth outcomes: A cross-sectional study in Malawi' has been provisionally accepted for publication in PLOS Global Public Health.

Best regards,

Gauri Desai, PhD

Academic Editor

The authors have done a good job of addressing the reviewer comments.